# Statistical Characteristics of the Response of Sea Surface Temperatures to Westward Typhoons in the South China Sea

**Zhaoyue Ma** [1], **Yuanzhi Zhang** [1,2,*], **Renhao Wu** [3] and **Rong Na** [4]

1 School of Marine Science, Nanjing University of Information Science and Technology, Nanjing 210044, China; mazy@nuist.edu.cn
2 Institute of Asia-Pacific Studies, Faculty of Social Sciences, Chinese University of Hong Kong, Hong Kong 999777, China
3 School of Atmospheric Sciences, Sun Yat-Sen University and Southern Marine Science and Engineering Guangdong Laboratory (Zhuhai), Zhuhai 519082, China; wurenhao@mail.sysu.edu.cn
4 College of Oceanic and Atmospheric Sciences, Ocean University of China, Qingdao 266100, China; nr@stu.ouc.edu.cn
* Correspondence: yuanzhizhang@cuhk.edu.hk; Tel.: +86-1888-885-3470

**Abstract:** The strong interaction between a typhoon and ocean air is one of the most important forms of typhoon and sea air interaction. In this paper, the daily mean sea surface temperature (SST) data of Advanced Microwave Scanning Radiometer for Earth Observation System (EOS) (AMSR-E) are used to analyze the reduction in SST caused by 30 westward typhoons from 1998 to 2018. The findings reveal that 20 typhoons exerted obvious SST cooling areas. Moreover, 97.5% of the cooling locations appeared near and on the right side of the path, while only one appeared on the left side of the path. The decrease in SST generally lasted 6–7 days. Over time, the cooling center continued to diffuse, and the SST gradually rose. The slope of the recovery curve was concentrated between 0.1 and 0.5.

**Keywords:** westward typhoon; South China Sea; typhoon influence; sea surface temperature

## 1. Introduction

A typhoon is a vortex characterized high temperature and a low pressure center structure formed in tropical ocean waters and the atmosphere, accompanied by strong air-sea interaction weather events. The phrase "westward typhoon" refers to a typhoon that moves westward after forming over the ocean surface near the Philippine Islands, enters the South China Sea through the Philippine Islands or the Bashi Channel south of Taiwan, and lands in the coastal areas of Southern China, Hainan Province or Vietnam [1]. Low pressure at the center of the typhoon and strong wind stress cause severe turbulent mixing and divergence in the upper ocean. In the form of wind entrainment, the typhoon deepens the depth of the mixed layer, and the typhoon-driven upwelling forces the upper sea-water to redistribute, resulting in a decrease in daily mean sea surface temperature (SST) [2]. At the same time, through the strong upwelling, nutrient-rich bottom water is pumped into the upper ocean, promoting the growth of phytoplankton over a large area and resulting in an increase in chlorophyll concentration. This phenomenon expands the primary productivity of the ocean and plays an essential role in the marine ecosystem [3–6]. Thus, investigating the influence of typhoons on the upper ocean in the study of marine environmental dynamics and ecology is of high significance.

As early as the 1950s, fishers used ship-based data to analyze changes in SST during tropical cyclone Connie, indicating that tropical cyclones forced SST to drop and finding that typhoons tended to move in warm sea areas [7]. An analysis of incomplete data provided preliminary findings that the SST drop caused by a typhoon is generally between 1~6 °C. Moreover, the prevalent cooling area is primarily located on the right side of the typhoon, and the deepening of the mixed layer corresponding to the decrease in SST also demonstrates the same right-oriented deviation [2,7–11]. For some time (5 days or

more) after the typhoon, the upper oceanic response to the typhoon continues, and the decreased SST affects both the intensification of the typhoon and further heat transfer, forming a negative feedback loop [12–14]. According to the measured data, the depth of the mixed layer increased by 3–10 m after the typhoon [8,9]. Another important response to typhoons is a near-inertial oscillation phenomenon in the upper ocean, which may cause the near-inertial shock gravity internal waves and strong flow that can last for several days [10]. However, the incompleteness of the observation data precludes a comprehensive analysis of the impact of typhoons on the upper ocean. Consequently, conducting a more in-depth and specific comparison and analysis of the change principle and mechanism is challenging.

The development of observation technology and instruments has made it possible for scholars to observe typhoons more comprehensively and in more detail. The results show that the intensity of the SST decrease is related to the depth of the mixed layer. In other words, the greater the extent of sea surface cooling, the greater the change in the depth of the mixed layer. In specific terms, a shallower initial mixing layer depth before a typhoon passes through an area leads to a greater cooling range for sea surface temperature. Furthermore, the mixing layer depth can deepen by tens to hundreds of meters [15,16], and the reduction of SST after a typhoon can last for up to 10 days [17]. The main reason for the variation in SST and mixing layer depth is that the mixing layer pulls in the lower cold water [2,15]. In addition, various mixing processes, including turbulence, upwelling and gales, and wave breaking also lead to a decline in SST. Liu et al. analyzed the response of the upper ocean to a passing tropical cyclone using Argo buoy observation data [14], finding that when the cyclone passed, the mixing layer depth observation profile can be deepened by 60 m. Additional findings revealed that the mixing layer depth changed significantly within 5 days following the cyclone, and the mixed layer temperature dropped, with maximum temperature falling by 5 °C, an average drop of 1.1 °C [14]. Although the temperature and salinity of the mixed layer decreased, the distribution of the two was completely different: the decrease in temperature has an obvious "right-bias", while the salinity is symmetrically distributed on both sides of the typhoon path [18–21].

At the same time, decreasing SST is also related to the pressure gradient, speed of the typhoon's motion, rain, and radiative processes. Additionally, the SST response to a typhoon is affected by the marine environment and structure [22–26]. Studies by Sun and others showed that, after a typhoon, the cyclone circulation appeared in the flow field, and the SST decreased significantly, with a range of decrease of about 2 °C, lasting for at least one week; moreover, the response area extended 300 km [27]. Statistical analysis shows that 70% of typhoons cause significant sea surface cooling (cooling ≥2 °C). In such cases, most of the maximum cooling from typhoons occurs on the right side of the typhoon track and is primarily concentrated in a range of about 100 km on either side of the typhoon's path [28]. Fu's statistical study on the impact of typhoons in the Northwest Pacific Ocean on the ocean SST found that the sustained response time of SST is 3–10 days after the typhoon passes [29]. Dong et al., analyzing the impact of Typhoon Megi on the South China Sea, found that a decrease of SST appeared on the left side of the typhoon track and caused upwelling [30]. The existence of cold and warm vortices on both sides of the typhoon path has a significant impact on the reduction in SST. If there is a strong cold vortex on the left side of the track, this will cause the typhoon's left side to cool down more significantly than the right side [28]. The combination of the existence of the cold vortex and the landing process caused the cooling effect on the right side of the typhoon path to be less obvious than the cooling effect on the left side [21,31].

Since most of the South China Sea is located in a tropical geographic region, the high evaporation leads to stratification of seawater that hinders the transportation of nutrients from the subsurface to the surface; thus the upper ocean water maintains an oligotrophic status throughout the year [32,33]. After a typhoon, however, SST decreased significantly, mixed layer depth increases, and sea surface chlorophyll concentration increased significantly [32–37]. The typhoon-caused prodigious mixing of ocean waters leads to an increase

in energy and material exchange between the upper ocean and the subsurface sea-water, exerting significant economic and social effects on the local offshore fishery, marine transportation, marine operations, and other industrial departments. Therefore, the response of the upper ocean of the South China Sea to typhoons has become a research hotspot in recent years.

Regarding research on the impact of typhoons on the upper ocean, it is mostly concentrated in the Northwest Pacific, Atlantic Ocean and other oceans [2,38–40]. During the typhoon passing, it is hard to collect data due to the influence of the harsher climatic environment. Therefore, most studies are based on the analysis of typical typhoon cases [11,13,41–44]. However, there are few studies on the statistical characteristics of the SST response of the upper ocean to typhoons in the South China Sea, especially to westward typhoons. SST is an important comprehensive characterization parameter of ocean thermal-dynamic process and air-sea interaction. SST plays an essential role in the exchange of energy and momentum between the ocean and the atmosphere, and is also a core element of air-sea interaction and ocean climate change [45]. Therefore, investigating the statistical characteristics of the response of the SST westward typhoons in the South China Sea can help understand the impact of the westward typhoon on the SST of the South China Sea, which is conducive to further research on marine environmental dynamics and ecology.

## 2. Materials and Methods

### 2.1. Study Area and Typhoon Data

The South China Sea is a semi closed basin with an average water depth of about 200 m and the deepest depth of the basin is more than 5 km (shown in Figure 1). It is the largest marginal sea on the east coast of Asia in the Western Pacific Ocean and one of the regions with the most frequent tropical cyclones and typhoons, with an average of 10.3 typhoons passing through each year [46,47]. As much of the South China Sea is located in the tropics, where high evaporation causes seawater to layer and hinders the transport of nutrient salts from the bottom up, the upper sea water is in a nutrient-innocencing state throughout the year [48,49].

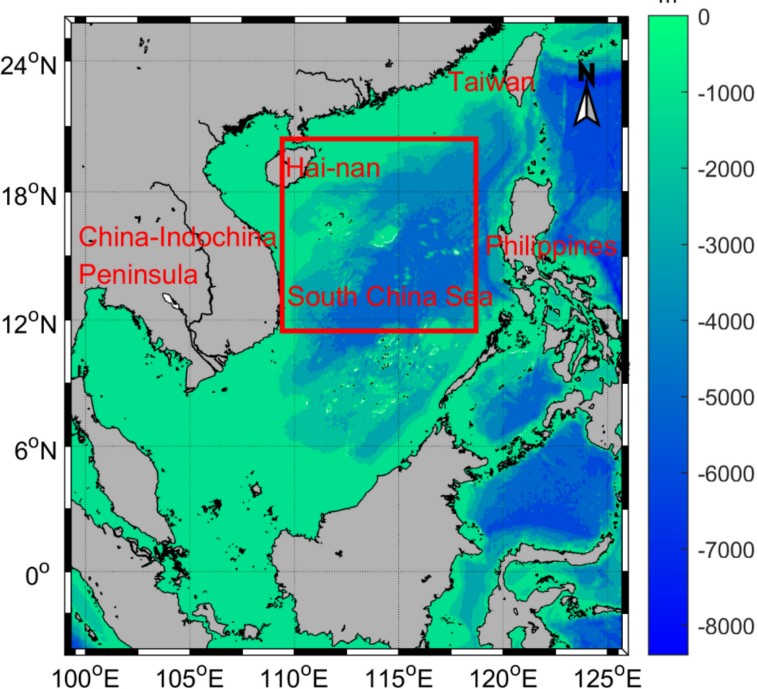

**Figure 1.** A bathymetric map of the South China Sea: The red box in the picture is the high frequency area where the temperature drops.

According to the revised national standard "tropical cyclone classification" issued by China Meteorological Administration in 2006, tropical cyclone can be divided into six grades according to the maximum average wind speed near the bottom center: tropical depression (TD), tropical storm (TS), severe tropical storm (STS), typhoon (TY), strong typhoon (STY) and super typhoon (Super TY) [50]. The typhoon mentioned in this paper refers to the tropical cyclone whose maximum average wind speed is more than 20 m/s (i.e., the level is above TS) near the center.

Using the best path provided by the tropical cyclone data center of China Meteorological Administration, we statistically analyzed the impact of 30 westward typhoons on the upper ocean from 1998 to 2018. Figure 2 is the path of selected typhoons in the study and Table 1 shows the basic information of each typhoon.

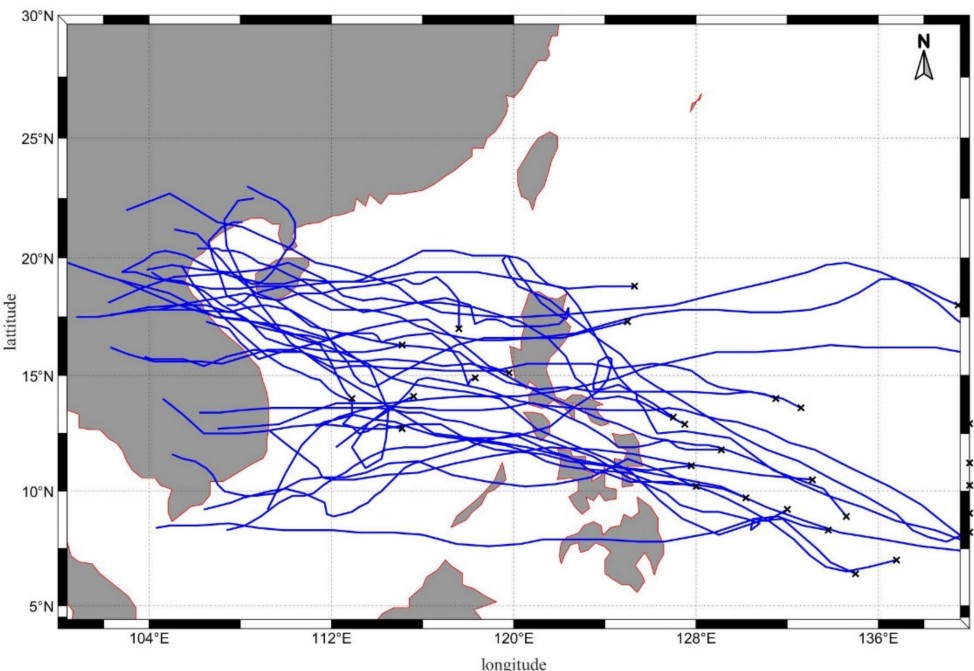

**Figure 2.** The best path of westward typhoon crossing the South China Sea from 1998 to 2018.

### 2.2. Data Sources and Processing Methods

The topographic data of the South China Sea comes from ETOPO1 global topographic and water depth data. It is divided into two versions, the Ice Surface and Bedrock, which are basically the same. The difference is that when dealing with Antarctica and Greenland, the former gives the elevation with the ice sheet added, while the latter gives the elevation of the bedrock. The download address for Bedrock version is as follows https://www.ngdc.noaa.gov/mgg/global/relief/ETOPO1/data/bedrock/ (accessed on 1 March 2021).

Traditional infrared and visible light remote sensing methods are susceptible to cloud, rain and other complex weather phenomena, making it difficult to observe and study the low-level structure of typhoons. Microwave remote sensing can be observed day and night, and it can penetrate clouds and fog with little influence, making its application in extreme weather such as typhoon conditions unmatched by other types of sensors [45]. It is an effective way to continuously observe the ocean surface. This article uses Optimally Interpolated (OI) SST daily products (http://data.remss.com/ (accessed on 1 March 2021)) provided by Remote Sensing Systems (REMSS). The data in this data set before June 2002 (1998–2002) only includes Measuring Mission (TRMM) Microwave Imager (TMI) data, the spatial span of the data is 40°N–40°S, its time resolution is 1 day, the spatial resolution is 0.25° × 0.25°. The SST data set after June 2002 includes TMI, Advanced Microwave Scanning Radiometer for EOS (AMSR-E), The Advanced Microwave Scanning Radiometer 2 (AMSR-2), WindSat, or GPM (The Global Precipitation Measurement) Microwave Imager

(GMI) daily SST data, its time resolution is 1 day, the spatial resolution is 0.25° × 0.25°, data covers the world. This data set removes the influence of daytime sea surface temperature warming and the SST data of rainwater pollution, and meets the research requirements of sea surface temperature changes after the typhoon passes. The typhoon data in this study are from the tropical cyclone data center of China Meteorological Administration (Shanghai Typhoon Research Institute) http://tcdata.typhoon.gov.cn (accessed on 1 March 2021), in which the data were recorded the typhoon center location, central pressure value and maximum wind speed every 6 h [51].

**Table 1.** Basic information of westward typhoons passing through the South China Sea.

| Times | Names | Study Area | Typhoon Rank | Maximum Wind Velocity (m·s⁻¹) |
|---|---|---|---|---|
| 1998/12 | Faith | 5°~25°N, 104°~125°E | STS | 30 |
| 1999/10 | Eve | 5°~25°N, 104°~125°E | TS | 23 |
| 2000/11 | Rumbia | 9°~9.5°N, 109°~110°E | TY | 40 |
| 2000/09 | Wukong | 19°~20°N, 115°~116°E | TS | 23 |
| 2001/12 | Kaijiki | 5°~25°N, 104°~125°E | TS | 20 |
| 2001/11 | Lingling | 13°~15°N, 113°~116°E | STY | 50 |
| 2003/08 | Krovanh | 18.5°~19.6°N, 114°~115°E | TY | 40 |
| 2004/11 | Muifa | 5°~25°N, 104°~125°E | TY | 40 |
| 2005/09 | Damrey | 19°~21°N, 112°~113°E | STY | 50 |
| 2005/10 | Kai-tak | 12.5°~15.1°N, 111°~113°E | TY | 40 |
| 2006/09 | Xangsane | 16°~17°N, 109°~111°E | STY | 50 |
| 2007/09 | Lekima | 5°~25°N, 104°~125°E | TY | 33 |
| 2008/09 | Mekkhala | 16.5°~17.5°N, 110°~111°E | TS | 23 |
| 2009/09 | Parma | 20°~22°N, 118°~120°E16°~17°N, 110°~111°E | Super TY | 55 |
| 2009/09 | Ketsana | 16°~17°N, 110°~111°E | TY | 40 |
| 2009/10 | Mirinae | 14°~15°N, 111°~113°E | TY | 40 |
| 2010/07 | Conson | 17.5°~18°N, 111.5°~112°E | TY | 35 |
| 2010/08 | Mindulle | 18°~19°N, 107°~108°E | STS | 30 |
| 2011/07 | Nockten | 18°~19°N, 111.5°~112.5°E | STS | 28 |
| 2011/09 | Nalgae | 17°~18.5°N, 115°~116°E | Super TY | 50 |
| 2012/10 | Son-tinh | 5°~25°N, 104°~125°E | STY | 45 |
| 2013/09 | Wutip | 18°~18.5°N, 112°~112.8°E | STY | 45 |
| 2013/10 | Nari | 5°~25°N, 104°~125°E | STY | 42 |
| 2013/11 | Haiyan | 5°~25°N, 104°~125°E | Super TY | 78 |
| 2014/12 | Hagutip | 15°~16°N, 115°~116°E | Super TY | 68 |
| 2016/12 | Nock-ten | 16.5°~18°N, 117°~119°E | Super TY | 62 |
| 2017/11 | Damrey | 13°~14.3°N, 113°~114°E | STY | 42 |
| 2017/12 | Tembin | 5°~25°N, 104°~125°E | TY | 38 |
| 2018/07 | Son-tinh | 18°~19°N, 107°~108°E | TS | 25 |
| 2018/11 | Usagi | 5°~25°N, 104°~125°E | TY | 33 |

To study the influence of typhoons on the SST of the upper ocean more intuitively, we deal with the temperature data as follows: First, to avoid abnormal SST in the single day before a typhoon, reduce the calculation error, the SST data of the 2 days before the occurrence of the typhoon in the South China Sea (5°–25°N, 100–140°E) is averaged to obtain the average SST mean value. Then, subtract the value of mean1 from the SST of 1-6 days after the typhoon passed to obtain sea surface temperature anomaly (SSTA) values, and compare them respectively. The reason for choosing 1–6 days after the typhoon as the research time is that a large number of studies have shown that the impact of the typhoon on the upper ocean lasts for about a week, but it is not a fixed value [2,7–22], so the actual statistics will adjust the number of days according to the actual situation. In the study, all typhoon tracks and remote sensing data were selected and mapped by MATLAB software.

## 3. Results

### 3.1. Statistics Regading the Influence of Westward Typhoon on SST

The cyclonic wind field generated by a typhoon and the associated extensive mixing of seawater has a profound impact on upper ocean dynamics. After the typhoon passes through, under the force of the wind field, the ocean is characterized by Ekman transport from the center of the cyclone to the outside, resulting in the outward transport of the surface water and a reduction in sea surface height. Meanwhile, the Ekman pumping effect cause the cold water in the lower part of the thermocline to rise to the mixing layer, resulting in vigorous turbulent mixing that leads to a decrease in SST. To date, although many scholars have studied observational statistics and the simulation of SST changes caused by typhoons in the South China Sea, most studies have been limited to individual cases [11,52]. Accordingly, we study the impact of typhoons on the upper ocean of the South China Sea by statistical analysis of 30 westward typhoons (TY and TS) from 1998 to 2018 to yield a more comprehensive understanding of the characteristics of SST changes caused by typhoons in the region. Following the lead of previous studies, this study defines a decrease in SST $\geq 2$ °C as obvious cooling [12,17,28]. Table 2 displays the statistical results.

**Table 2.** Statistical information of typhoon track cooling.

| Typhoon Number | Names | Minimum Pressure (hPa) | Maximum Wind Velocity (m·s⁻¹) | Cooling Position [1] | Maximum Cooling Value (°C) | Standard Deviation |
|---|---|---|---|---|---|---|
| 199824 | Faith | 980 | 30 | — | 1.02 | 0.5377 |
| 199927 | Eve | 990 | 23 | — | 0.83 | 0.3626 |
| 200023 | Wukong | 960 | 40 | Right | 2.62 | 0.6850 |
| 200033 | Rumbia | 990 | 23 | Right | 1.57 | 0.4895 |
| 200127 | Lingling | 996 | 20 | Right | 6.45 | 0.9031 |
| 200130 | Kaijiki | 940 | 50 | — | 1.35 | 0.5147 |
| 200312 | Krovanh | 960 | 40 | Right | 3.15 | 0.6067 |
| 200439 | Muifa | 960 | 40 | — | 1.78 | 0.6786 |
| 200517 | Damrey | 940 | 50 | Right | 4.80 | 0.7916 |
| 200522 | Kai-tak | 960 | 40 | Near | 7.88 | 1.2216 |
| 200618 | Xangsane | 945 | 50 | Right | 5.55 | 0.8689 |
| 200716 | Lekima | 975 | 33 | — | 1.65 | 0.8208 |
| 200820 | Mekkhala | 990 | 23 | Right | 3.37 | 0.7045 |
| 200919 | Parma | 960 | 55 | Left | 3.67 | 0.8230 |
|  |  |  |  | Right | 3.98 | 0.8230 |
| 200917 | Ketsana | 960 | 40 | Right | 6.45 | 1.2693 |
| 200923 | Mirinae | 940 | 40 | Right | 3.30 | 0.6582 |
| 201003 | Conson | 980 | 35 | Right | 3.23 | 0.9266 |
| 201006 | Mindulle | 970 | 30 | Right | 3.45 | 0.5437 |
| 201110 | Nockten | 940 | 28 | Right | 2.10 | 0.9369 |
| 201122 | Nalgae | 985 | 50 | Right | 6.08 | 0.9512 |
| 201224 | Son-tinh | 950 | 45 | — | 1.65 | 0.5289 |
| 201320 | Wutip | 940 | 45 | Right | 4.05 | 0.7445 |
| 201324 | Nari | 960 | 42 | — | 1.45 | 0.8336 |
| 201331 | Haiyan | 955 | 78 | — | 1.25 | 0.5308 |
| 201422 | Hagutip | 920 | 68 | Right | 2.93 | 0.6367 |
| 201630 | Nock-ten | 915 | 62 | Right | 3.07 | 0.6837 |
| 201728 | Damrey | 955 | 42 | Right | 3.53 | 0.8039 |
| 201733 | Tembin | 965 | 38 | — | 1.24 | 0.5658 |
| 201811 | Son-tinh | 990 | 25 | Near | 3.75 | 0.5311 |
| 201833 | Usagi | 980 | 33 | — | 0.97 | 0.4177 |

[1] "Right" represents the cooling center on the right side of the path; "Left" represents the cooling center on the left side of the path; "Near" represents the cooling center near the path; "—"represents there was no significant cooling.

The statistics show that 20 of the 30 westward typhoons from 1998 to 2018 invoked significant cooling, the occurrence of cooling typhoons accounted for 66.67% of the total westbound typhoons. The cooling range was 2–8 °C, and the maximum cooling was

7.88 °C. Most of the cooling ranges were between 3–4 °C. The standard deviation of the SSTA of most typhoons is less than 1.00, with two exceptions-the standard deviation of the 200522 typhoon Kai-tak is 1.2216, and the standard deviation of the 200917 typhoon Ketsana is 1.2693. Clearly, this has no effect on the subsequent analysis of the results.

The reasons for the changes in SST after the typhoon, the most intuitive influencing factors are the maximum wind speed of the typhoon and the low-pressure central pressure. The correlation analysis between the two and the maximum cooling value indicated (see Figure 3) that a maximum cooling value has no obvious correlation with the low-pressure central pressure and the maximum wind speed.

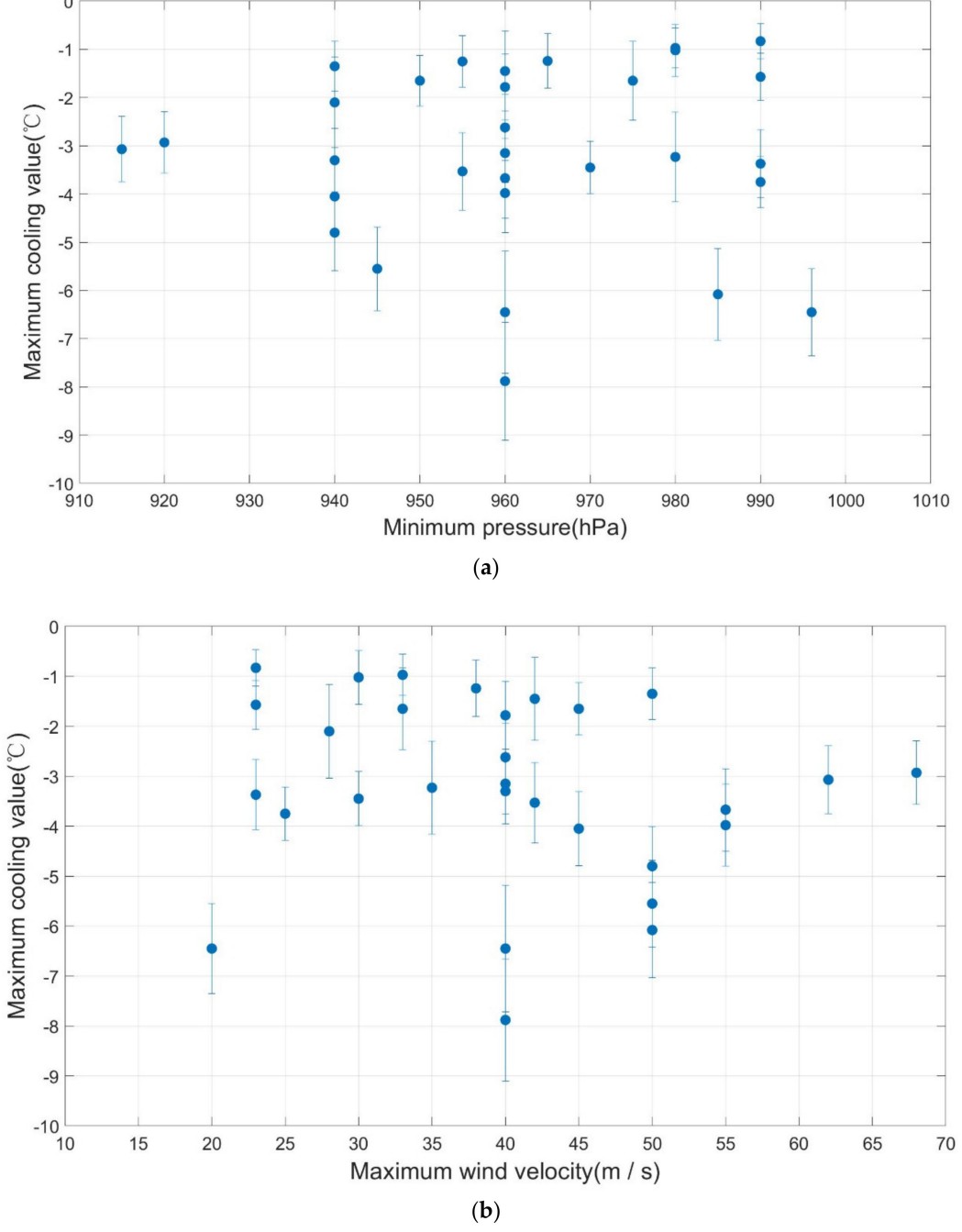

**Figure 3.** Correlation of maximum cooling value with low pressure center pressure and maximum wind velocity: (**a**) Description low pressure center; (**b**) Description maximum wind velocity.

Taking 2011's No.22 typhoon (Nalgae) as an example (Figure 4), we can see the obvious cooling areas that appeared on the surface of the South China Sea after the typhoon passed through. On the first day after the typhoon, the cooling areas were concentrated, and the cooling range was at the greatest. With the passage of time, the cooling center gradually diffused while the range of temperature reduction gradually decreased. The SST before the typhoon was basically restored by the sixth day. Furthermore, the statistics reflecting the SST change range of obvious cooling after the westward typhoon from 1998 to 2018 (as shown in Figure 5), demonstrate a certain similarity in the range of temperature rise in the six days after typhoon transit.

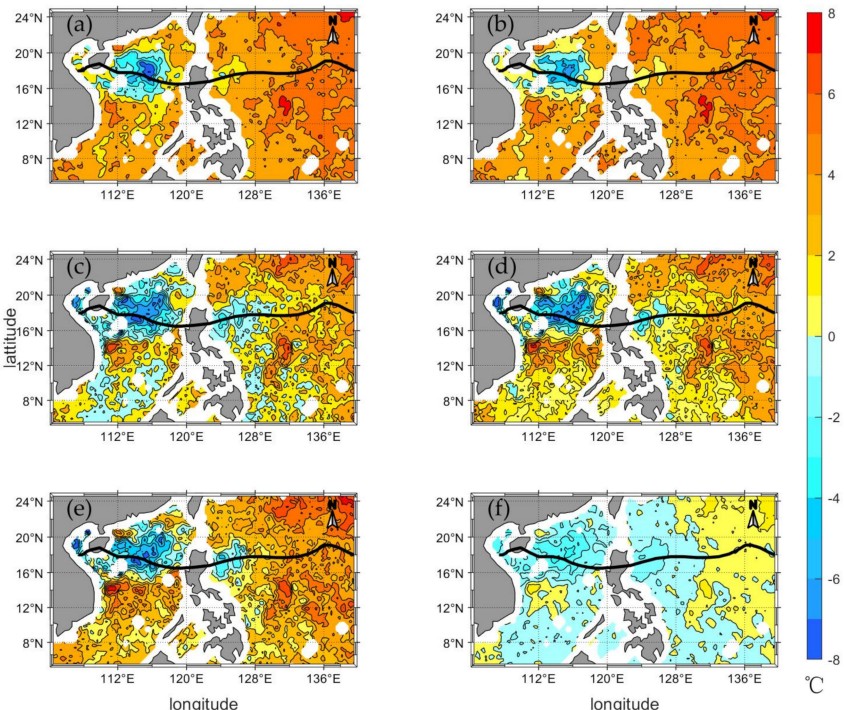

**Figure 4.** Variation of SST in the South China Sea in six days after typhoon Nalgae's transit: (**a**–**f**) corresponds to the SST change of 1–6 days after the typhoon.

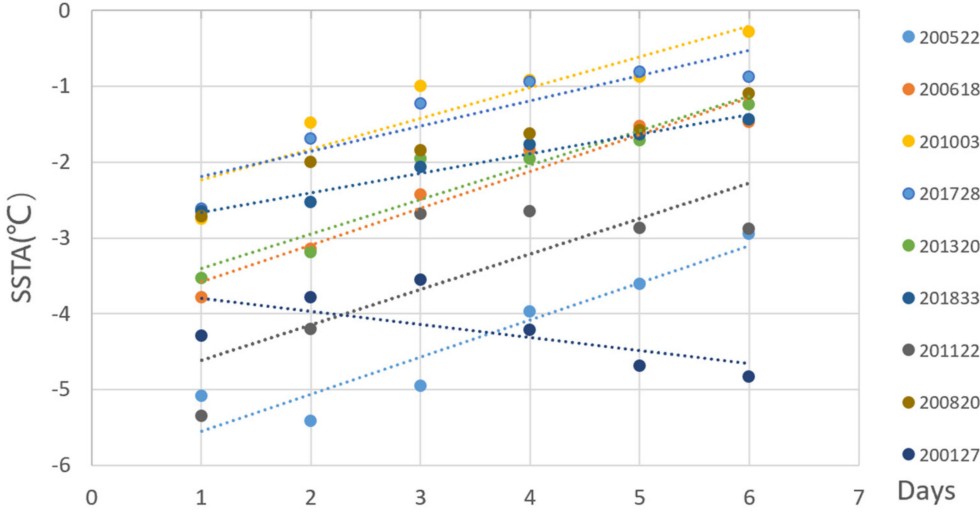

**Figure 5.** Rising trend line of SST after westward typhoon: The legend part is the typhoon number.

Further analysis of the slope of the SST recovery trend line reveal that the SST recovery slope was between 0.1–0.5 for the 20 westward typhoon that caused obvious cooling as

shown in Figure 6. The only exception is typhoon (Lingling) on the 27th in 2001, with a temperature slope of −0.17.

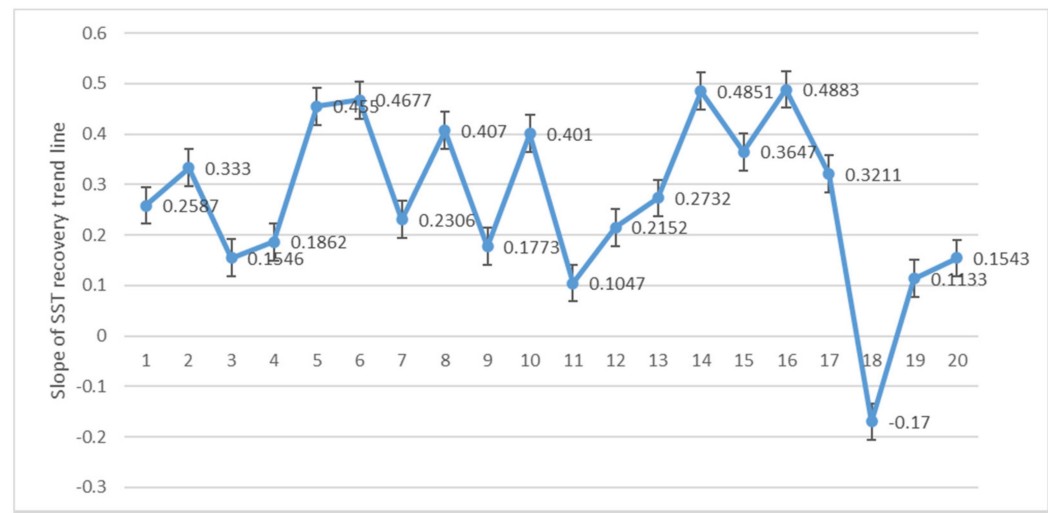

**Figure 6.** Slope of sea surface temperature recovery trend line after Typhoon.

Typhoon Lingling formed in the sea off the eastern coastal line of the Philippines at 0:00 on 6 November 2001, as a tropical depression, strengthening into a typhoon at 18:00 on that day. After passing through the South China Sea, the SST on the right side of the path decreased significantly, and the maximum cooling range was 6.45 °C. An obvious cold vortex appeared on the right side of the typhoon track, and the response lasted for 12 days, as shown in Figure 7.

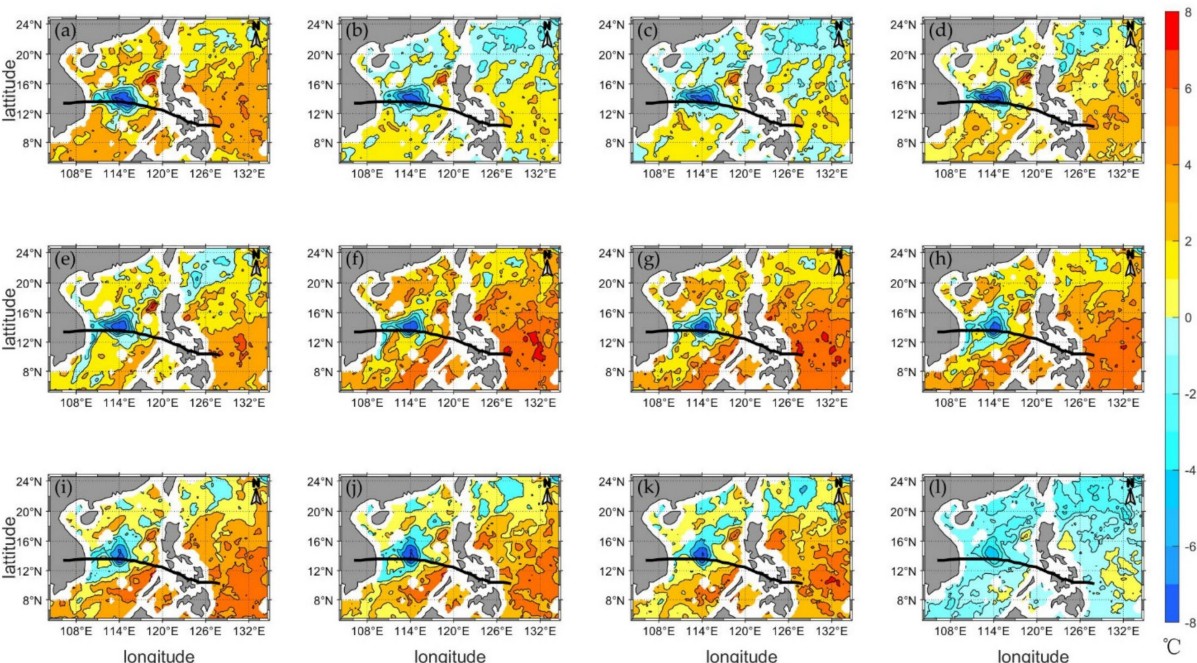

**Figure 7.** (**a–l**) shows changes of SST in 1–12 days after typhoon Lingling passed through.

### 3.2. Statistics Related to Cooling Position after Westward Typhoons

The statistics show that the westward typhoons in the South China Sea can be divided into two categories according to the change in SST after transit: one is no obvious

cooling (cooling ≤2 °C), as shown in Figure 8, typhoon Usagi, which was generated on 17 November 2018; the other is the obvious drop in SST. Then, it can be divided into three categories according to the location of the cooling: (1) the cooling position appeared on the right side of typhoon track, such as Mirinae–the 23rd typhoon in 2009; (2) the cooling position appeared on the left side of the typhoon track, like Parma, the 19th typhoon in 2009; and (3) the center of the cooling position appeared near the typhoon track, represented by Typhoon Kai-tak in 2005.

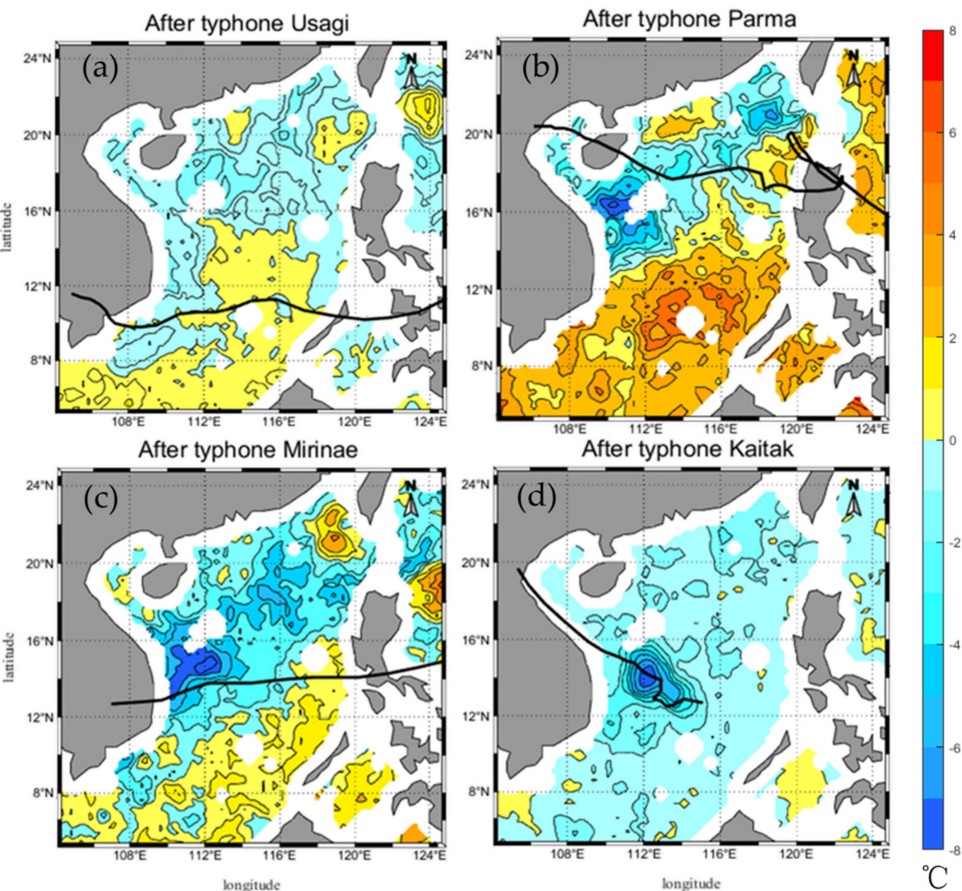

**Figure 8.** Types of SST changes caused by westward typhoon: (**a**) Describes the change in sea surface temperature one day after Typhoon Usagi; (**b**) Describes SSTA one day after Typhoon Parma; (**c**) Describes SSTA one day after Typhoon Mirine; (**d**) Describes the changes in sea surface temperature on the second day after Typhoon Kaitak.

According to the statistics for typhoon track cooling (Table 2), 20 of the 30 westward typhoons experienced obvious SST cooling after passing through, 17 of which appeared on the right side of the typhoon track, while two appeared near the central track, and only one appeared on the left side of the track. However, no significant cooling on the surface of the South China Sea was observed after the other 10 typhoons under consideration passed through.

## 4. Discussion

### 4.1. Response of South China Sea Surface Temperature to a Westward Typhoon

A typhoon is usually accompanied by enormous wind stress and anticyclonic structure (the vertical movement of the typhoon eye) in the center of a low-pressure area. The joint action of strong wind and anticyclone tendency will cause strong mixing and turbulence in the upper ocean layer and will deepen the mixing layer depth. The rise of subsurface water caused by Ekman pumping redistributes the sea surface water and leads to a decrease in SST [2]. Through the processing and analysis of microwave remote sensing data, we found

that a 66.67% probability for 30 westerly typhoons that passed through the South China Sea from 1998 to 2018 where a typhoon caused the SST to drop significantly. The cooling range was between 2–8 °C, and the cooling duration could extend to 6–7 days. That said, the cooling caused by Typhoon Lingling lasted 12 days. The location of cooling usually appeared on the right side of the typhoon track. Among the 20 typhoons with obvious cooling, 17 cooling centers appeared on the right side of the typhoon track, accounting for 84.21%, while the cooling centers of two typhoons—Son-Tinh (No.11 in 2018) and Kai-Tak (No.22 in 2005) appeared in the middle of the track; and only one Typhoon Parma (No.19 in 2009) had a cooling center on the left side of the typhoon track.

The location of typhoon-caused SST cooling relative to the typhoon track is the result of multiple factors, including asymmetry of typhoon wind stress, horizontal flow, the depth distribution of the mixed layer on both sides of the typhoon track, isotherm distribution, and the vertical gradient of sea surface temperature [22]. In the northern hemisphere, the asymmetry of wind stress and advection of a typhoon make the SST decrease to the right of the typhoon track. Compared with other types of typhoons, westward typhoons have a relatively uniform impact on the upper ocean because of their simple path and clear developmental process, and the probability of exception occurring is relatively low.

### 4.2. Reasons for the Long Duration of Typhoon Lingling

As shown in Figure 7, after the passage of Typhoon Lingling, an obvious cooling center appeared on the right side of the typhoon track. In this case study, the slope of the temperature rise curve differed from that of most typhoons, showing negative growth (as shown in Figures 5 and 6). It can be seen from Figure 9 that the temperature in the cooling area did not rise in the six days after typhoon Lingling; on the contrary, SST continued to decline on the third day, and only after reaching the lowest temperature on the seventh day did it show a rising trend. The slope of the temperature rise curve was 0.2234, ranging from 0.1–0.5.

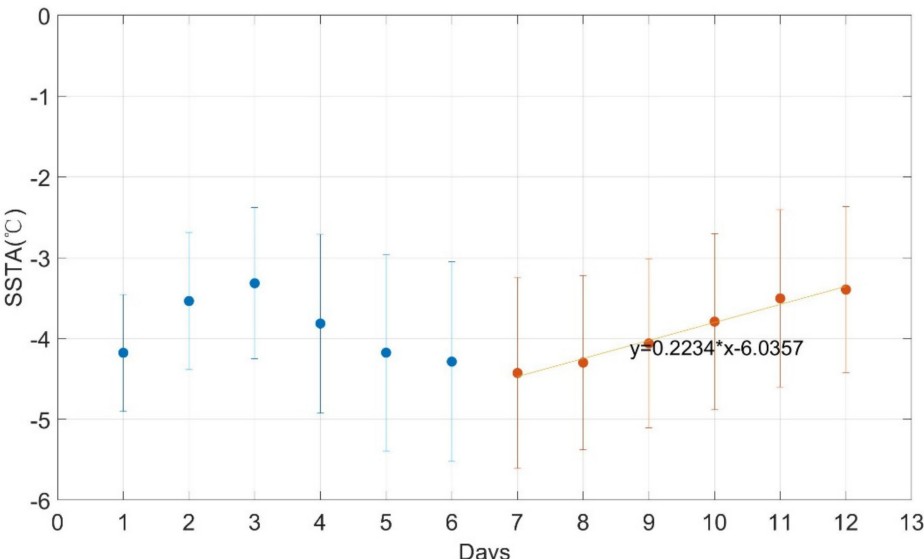

**Figure 9.** Variation trend of sea surface temperature after typhoon Lingling passes through.

First, according to the available typhoon data, no other typhoon appeared in the central part of the South China Sea a week before or after Lingling. Yang et al. studied the surface height data of the South China Sea and found an obvious cold vortex near the typhoon track one week before the formation of Typhoon Lingling. After the typhoon, the scope of the cold vortex expanded and the cooling intensity was greatly strengthened [28]. Therefore, we infer that the change in SST after the typhoon passed by the cold vortex differed from that of other westbound typhoons, and the cooling time and temperature recovery lasted longer.

### 4.3. The Cooling Center Appearing Near the Path

Among the 20 westward typhoons with obvious sea surface cooling after transit, only two displayed cooling centers near the typhoon track without an obvious tendency. It can be seen from Figure 10 that at 18:00 on 18 July 2018, the path of typhoon Son-Tinh changed after its landing in Vietnam. At 6:00 on 21 July, when the typhoon intensity dropped to the level of a tropical depression, the typhoon center returned to the Beibu Gulf in the South China Sea, and gradually strengthened to the level of tropical storm. Two typhoons passed through the study area over a short period, and they caused upwelling in the area for a long time, resulting in the decrease in SST.

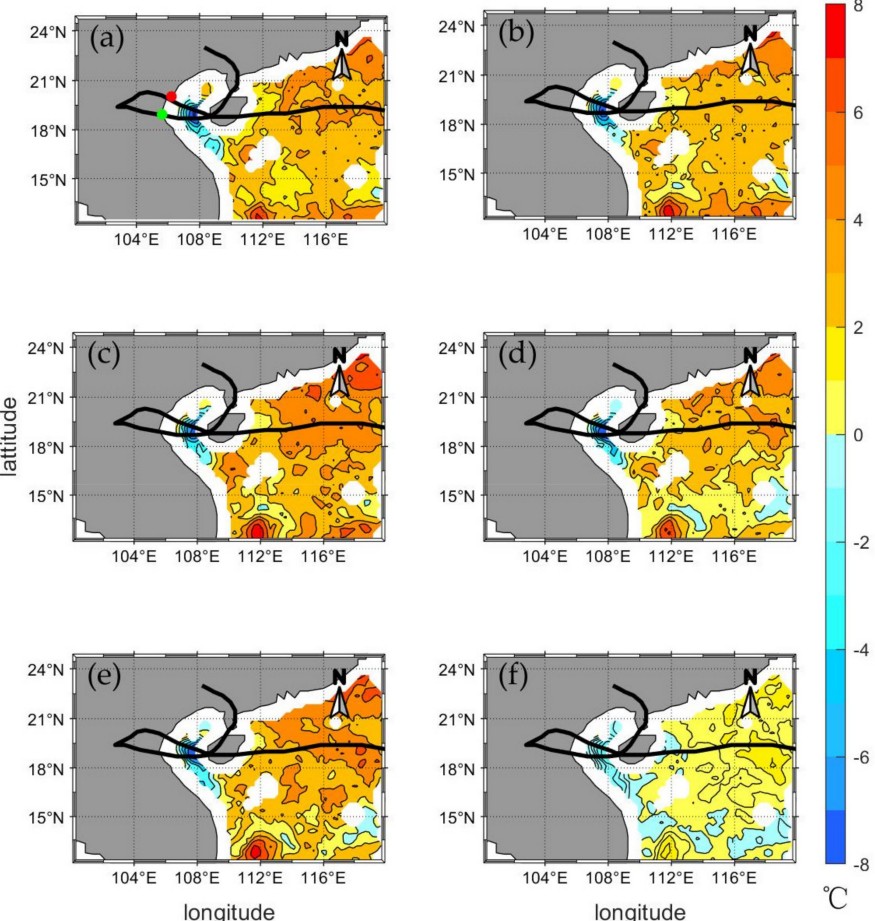

**Figure 10.** Temperature change after typhoon Son-tinh in 2018: (**a**) Describes the change of SSTA on the first day after the typhoon, the green dot represents the typhoon center at 18:00 on 18 July, the red dot represents the typhoon center at 6:00 on 21 July; (**b–f**) Describes the change of SSTA on 2–6 day after the typhoon.

Typhoon Kai-Tak formed at 18:00 on 27 October 2005. Figure 11 shows an obviously bent typhoon track in the western part of the South China Sea. Looking up the best path data for the typhoon, we found that the center of the typhoon moved slowly and stayed in the cooling area for 48 h (from 18:00 on 28 October to 18:00 on 30 October), while the maximum wind speed increased from 18 to 40 m/s. This typhoon's significant and extended winds in the region caused extensive turbulent mixing on the sea surface, resulting in the upwelling of subsurface water and an obvious decrease in SST.

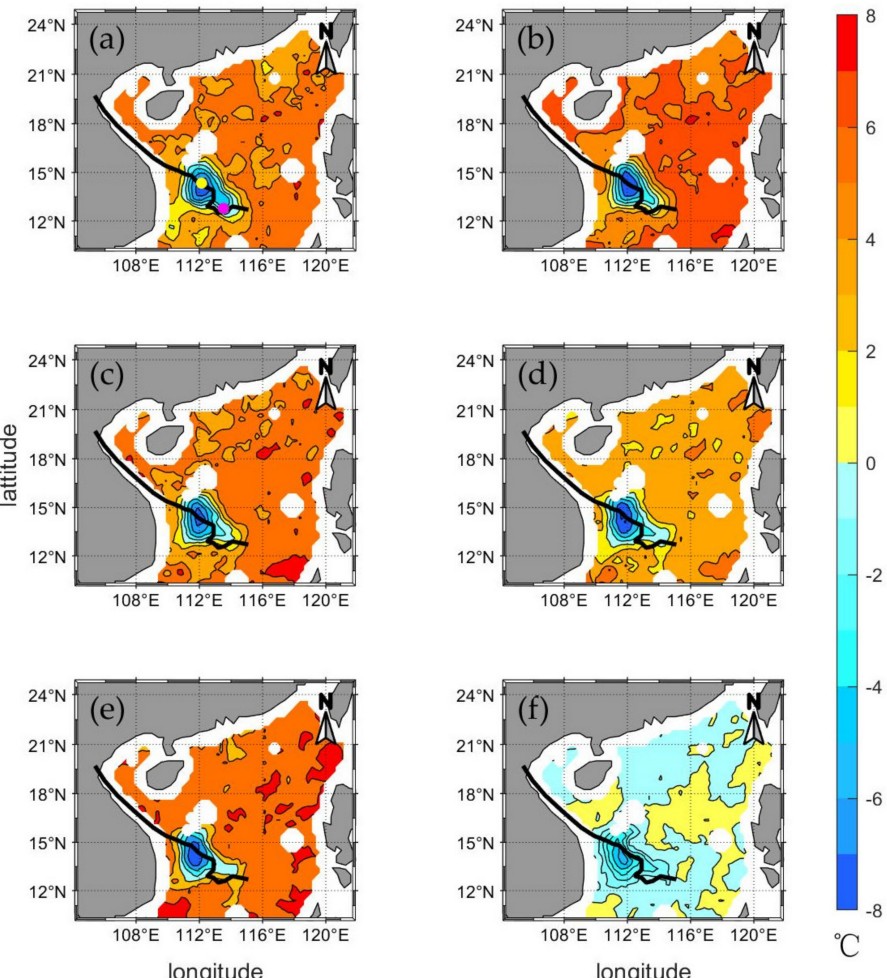

**Figure 11.** The track of typhoon Kai-tak and the temperature change after its passage in 2005: (**a**) Describes the change of SSTA on the first day after the typhoon. The magenta dot represents the typhoon center at 18:00 on 28 October, the yellow dot represents the typhoon center at 18:00 on 30 October; (**b**–**f**) Describes the changes in SSTA 2–6 days after the typhoon.

*4.4. The Cooling Center Appearing on the Left Side of the Path*

Typhoon Parma formed at 0:00 on 27 September 2009, in a tropical low off the eastern side of the Philippines and strengthened into a typhoon at 18:00 on 6 October. Figure 12 demonstrates that under the influence of a subtropical high, Parma made landfall again in the Philippines after crossing that body of land into the South China Sea (18:00 on 3 October), returned to the South China Sea again at 18:00 on 8 October, then proceeded westward through Hainan Island, finally landing in Vietnam at 6:00 on 14 October. Analysis of SST after Parma reveals that two regional cooling events in the South China Sea, one in the northeast of that body of water (20°~22°N, 118°~120°E), to the right of the typhoon track, and the other in the western part of the South China Sea (16°~17°N, 110°~111°E), on the left side of the typhoon's path.

Looking at the typhoon's path shows that before Typhoon Parma passed through the area, No. 17 Typhoon Ketsana (formed at 0:00 on 25 September, landed in Vietnam at 6:00 on 29 September, and died at 18:00 on 30 September) also passed through the region and caused cooling in the western part of the South China Sea, with a maximum temperature drop that reached 6.45 °C. Figure 13 shows the two typhoons in chronological order. Figure 13a–e reveals the emergence of a cooling center on the right side of the track after Typhoon Ketsana; the temperature gradually rises over time. However, the passage of Typhoon Parma not only slowed down the rising temperature trend in the region but also

further strengthened the intensity of the cold vortex, which led to a decrease in temperature and caused the duration of the cold vortex to last for two weeks.

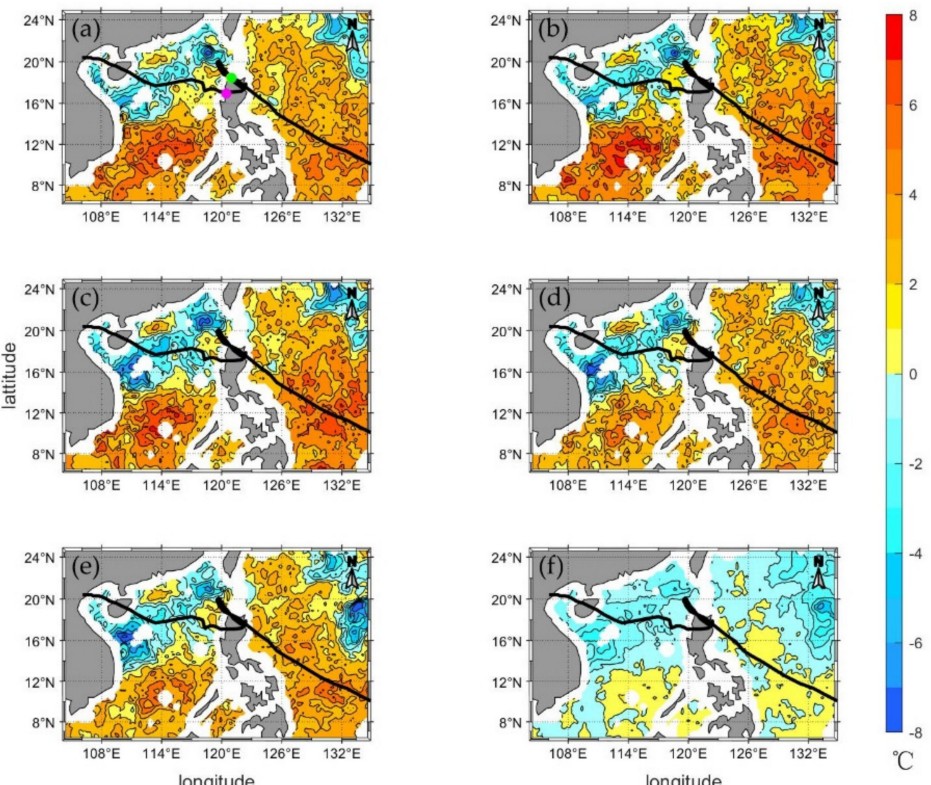

**Figure 12.** Track of Typhoon Parma and temperature change after its transit in 2009: (**a**) Describes the change of SSTA on the first day after the typhoon. The green dot represents the typhoon center at 18:00 on 3 October and 18:00 on the 6th; the magenta dot represents the typhoon center at 18:00 on 8 October; (**b**–**f**) Describes the changes in SSTA 2–6 days after the typhoon.

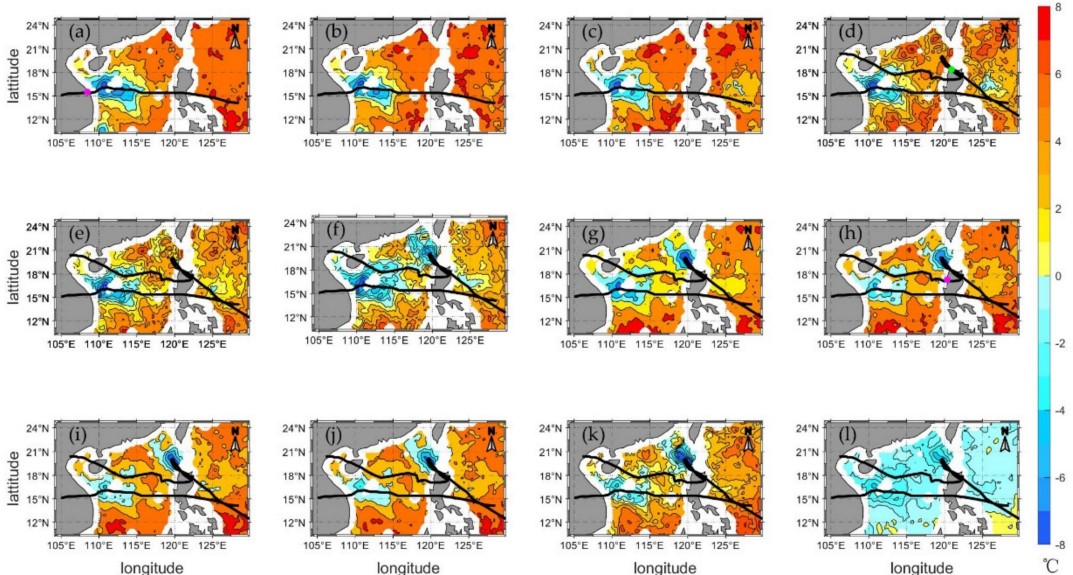

**Figure 13.** Surface temperature changes in the South China Sea after the passage of Typhoon Ketsana and Parma in 2009. (**a**) shows SSTA in the South China Sea on 30 September, (**b**–**l**) shows October 1st to 11th; (**a**–**e**) shows the separate track of typhoon Ketsana; (**f**–**l**) shows putting in the best path of Parma.

## 5. Conclusions

This paper presented an analysis of the impact of 30 westward typhoons on the upper ocean of the South China Sea during the 21 years from 1998 to 2018 from two aspects: SST changes and the location of cooling relative to the typhoon track. From the above statistical analysis, it is concluded that 66.67% of westward typhoons in that timeframe caused a significant decrease in SST ($\geq$2 °C) in the upper ocean of the South China Sea, and the range of cooling was between 2–8 °C. The standard deviation of most SSTAs is less than 1, which has no significant influence on the final result analysis. The maximum cooling range had no obvious correlation with the low-pressure central pressure and maximum wind speed. Among them, the typhoons considered, 87.5% (17.5) of the cooling areas were on the right side of the path, 10% (2) were near the path, and only 2.5% (0.5) were on the left side of the path. Due to the relatively simple track of a typical westward typhoon, the influence of the Coriolis force is significant. The asymmetry of wind stress and the effect of advection on typhoon lead to a decrease in SST to the right side of the typhoon track in the northern hemisphere.

With the passage of time, the cooling center continuously diffuses to the surrounding areas, allowing the temperature to rise gradually, and the slope of the recovery curve is between 0.1 and 0.5. Consequently, the cooling area generally returns to normal SST in 6–7 days. However, the presence of a cold vortex on the sea surface before a typhoon passes through will increase the cooling duration.

The response of the upper ocean to a typhoon is a very complex dynamic process, which is not only affected by the seabed topography and landform, but also by dynamic factors, including local wind and flow fields. This study focuses on statistical characteristics, analyzing the statistical results of the SST response to westward typhoons in the South China Sea. Further studies are still needed to investigate the specific dynamic process of the surface temperature response to typhoons in the South China Sea.

**Author Contributions:** Conceptualization, Y.Z. and Z.M.; methodology, R.W.; software, R.N.; validation, Z.M., R.N. and R.W.; formal analysis, Z.M.; investigation, Z.M.; resources, Y.Z.; data curation, Z.M.; writing—original draft preparation, Z.M.; writing—review and editing, Y.Z.; visualization, R.N.; supervision, Y.Z.; project administration, Y.Z.; funding acquisition, Y.Z. All authors have read and agreed to the published version of the manuscript.

**Funding:** This research was funded by the Marine Special Program of Jiangsu Province in China (JSZRHYKJ202007), the National Natural Science Foundation of China (U1901215), the Natural Scientific Foundation of Jiangsu Province (BK20181413), and the National Key Research and Development Program of China (Project Ref. No. 2018YFC1407200, 2018YFC1407203, and 2016YFC1402003).

**Data Availability Statement:** Publicly available datasets were analyzed in this study. This ETOPO1 global topographic and water depth data can be found here: [https://www.ngdc.noaa.gov/mgg/global/relief/ETOPO1/data/bedrock/] (accessed on 1 March 2021). This SST data can be found here: [http://data.remss.com/] (accessed on 1 March 2021). This typhoon data can be found here: [http://tcdata.typhoon.gov.cn] (accessed on 1 March 2021).

**Acknowledgments:** Satellite data and local data used in this study are highly appreciated. This research was funded by the Marine Special Program of Jiangsu Province in China (JSZRHYKJ202007), the National Natural Science Foundation of China (U1901215), the Natural Scientific Foundation of Jiangsu Province (BK20181413), and the National Key Research and Development Program of China (Project Ref. No. 2018YFC1407200, 2018YFC1407203, and 2016YFC1402003).

**Conflicts of Interest:** The authors declare no conflict of interest.

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
