# Peer review of "Statistical Characteristics of the Response of Sea Surface Temperatures to Westward Typhoons in the South China Sea"

_remotesensing, doi:10.3390/rs13050916_

Round 1

Reviewer 1 Report

Statistical characteristics of the response of the South China Sea to westward typhoons

by Zhaoyue Ma1, Yuanzhi Zhang, Renhao Wu and Rong Na

I find a number of issues that I would like the author to address before I could recommend this manuscript for publication.

Most importantly, the paper lacks any assessment of the accuracy of the SSTA . Without it one cannot really evaluate any statement about the SSTA and especially the recovery rate.

A discussion of the accuracy of their estimates is needed especially since the main data on which the finding of the study are based is the sea surface temperature obtained form microwave satellite sensors and the microwave signal is know to be distorted by presence of rain – a frequent occurrence during typhoons. Thus the accuracy of SST data might be insufficient to draw some of the conclusions of this study.

Specific remarks:

Title: the paper deals with the change in SST after a passage of a typhoon while the title suggested a responds of a much larger scope. It should probably be made more specific.

Number of cases: why are only 30 typhoons considered, what is the total number and what is the total number of westward typhoons over the study period

66 – what about rain, radiative processes

74 - moving speed of what?

88 – I don’t understand this sentence

93 – 103 this process will only matter if it is a frequent occurrence but there is on mention here how often these conditions form, moreover since only 20 typhoons from the period of 20 years are considered here this leaves impression these conditions are not very frequent and would not make big impact

124 - assume it is 10.3 typhoons per year but it should be spelled out here

149 – 159, this discussion of microwave SST data if very brief. It is the data set on which the whole study is based yet there is no discussion of the accuracy of the microwave SST, its limitations, why it was chosen above other data set (numerical analysis, drifter data, other satellite instruments). The problems with microwave SST in strong winds and rain – very much present in typhoons in particular would be a cause for concern. There is also the question of the skin SST measured by satellite instruments vs bulk SST more representative of the mix layer which I guess would be more appropriate for the purpose of this study.

153 – the 0.01C is a standard deviation of what ?

160- 166, meaning of the average temperature here is not clear, is this the average temperature of the whole basin? This needs to better defined.

181 – again is 30 westward typhoons the total number of westward typhoons between 1998 ad 2018, if not why these 30 were chosen

Table 2: and throughout the paper

the Maximum cooling rage is probably not the best term for this parameter as range implies two values (lower and upper limit) the the maximum value of cooling is just one number.

187 – cooling rate has not been defined but typically implies change over time so no consistent with the units of ‘%’

Figure 3, it seems that the cooling range=SSTA , just use SSTA and drop the cooling range. Also, the figure axes should probably be reversed as the minimum pressure and the max wind speed seem to the the independent variables that drive the SSTA.

Figure 4 could use the position of the typhoon’s center along the track on each day. That would provide addition information about the position of the center of the cooling and the center of the typhoon over time.

Figure 5 could use a legend to identify the tend lines for each typhoon

Figure 7 -13 , again it would be good to see where the center of the typhoon is in each panel

this is specially important in figure 13 which attempts to explain an interaction between two typhoons

Figure 8 and discussion, the snapshots in panels of Figure 8 should state the time after passage for each panel otherwise the claims made wrt TY Usagi in particular cannot be supported.

240 – I am not sure if there us a need for a new section here. Most of the ‘Discussion’ section seems to be really a continuation of the Results for the special cases.

Figure 9 , needs error bars (as do some of the earlier Figures too). Also putting tick labels at the bottom axis would make the plots more legible.

273 – I don’t see how the slope is ranging from 0.1 to 0.5

278 - ‘Some scholars’ should be replaced by names of those scholars. I guess we should infer that the presence of the cold vortex affected the SST response to the TY Lingling but if that’s the message then it should be clearly stated here.

286 – we cannot really see from the Figure when the path changed since there are no time stamps along the track so we don’t know when the TY is at any point in time. All tracks should indicate the position of TY center corresponding to the time of the panel

305 -306, we cannot see the subtropical high in Figure 12 nor how it influenced the TY Parma. Authors could add a synoptic chart showing the subtropical high are just describe the synoptic situation but in current form the statement is incorrect. Again having location of the TY center in the panels would be helpful.

335 – 352 the conclusion statements carry very little wight without any kind of error analysis.

Author Response

Dear Reviewer,

Thank you for your comments. We have revised and improved the manuscript as attached.

Yours sincerely,

Yuanzhi Zhang

Reviewer 2 Report

The manuscript titled “Statistical characteristics of the response of the South China Sea to westward typhoons” is interesting.

The authors should to pay more attention to figures and tables and better explain the analyzed data set.

Line 61: there is a typing error.

Figure 1: add the unit of measurement.

Lines 137 and 138: replace figure 1 with figure 2.

Figure 2: remove the sentence “The best path of westward typhoon crossing the South China Sea from 1998 to 2018” from the top of the figure 2 and add the highlighted area in figure 2 to figure 1.

Paragraph “2.2 Data Sources and Processing Methods”: indicate the error associated with these data

Line 162: explain why this time range, two days before the typhoon, was selected.

Line 164: explain why this time range, 1-6 days after the typhoon, was selected.

Lines from 181 to 183: better explain this sentence and the relationship to the values shown in Table 2 and Figure 3.

Figures from 4 to 11: add the unit of measurement.

Line 210: there is a typing error.

Figure 7: explain why this time range, “1-12 days after typhoon Lingling passed through”, was selected.

Lines from 249 to 251: in the “Paragraph “2.2 Data Sources and Processing Methods”, explain how many days after the typhoons were monitored to clarify these sentence “the cooling duration can reach 6-7 days” and “The cooling caused by typhoon Lingling is as long as 12 days”.

Line 278: there is a typing error.

Author Response

(The authors gave the same response as above.)

Reviewer 3 Report

My general comments and suggestions are as follows:

  1. INTRODUCTION

In this chapter the authors provide a good information about the main topics of their work. My suggestion is to improve this chapter in such a way as to describe in more detail the area defined in line 27 (insert suitable figure with the map of the area with all toponyms, e.g. China, Philippine Islands, South China Sea etc.) as well as the general characteristics of typhoons e.g. annual frequency and main routes. At the end of this chapter (after line 118) the main objectives of the paper should be described.

  1. MATERIALS AND METHODS

Line 124.  …. With an average of 10.3 typhoons … . Is it an average of 10.3 typhoons per year?

Figure 1. Insert suitable toponyms to recognize the area.

Lines 128-134. Please make a comment about “tropical cyclone classification” issued  by the China Meteorological Administration and the “Saffir-Simpson Hurricane Wind Scale” (SSHWS). SSHWS is based on 1-minute maximum sustained winds, starting from Category 1 (33 – 42 m/s) to Category 5 (≥ 70 m/s).

Line 133. The authors state:”.. whose maximum average wind speed is…”. What is a time interval of averaging?

Table 1.  According to this table neither one super typhoon have been recorded. But to my knowledge, typhoons Parma and Haiyan were super typhoons with maximum wind speeds 70 and 88 m/s respectively!? In general, the maximum wind speeds shown are not extreme, as defined by SSHWS for hurricanes!?

Line 153. Term “resolution accuracy” is used. This is not the correct term, because there is a difference between accuracy and resolution. Accuracy is how close a reported measurement is to the true value being measured. Resolution is the smallest change that can be measured!

Lines 154-155. Please correct this sentence and define what a deviation is!

  1. RESULTS

Line 208 and Figure 6. What does “slope of temperature recovery” means and does this value has its unit e.g. ° C/day!? What is presented on x-axes of the Figure 6? If my remark is correct, please correct all parts of the text from the abstract to the conclusion!

  1. DISCUSSION

Lines 242-243.  The authors state:” Typhoon is usually accompanied by huge wind stress and the anticyclone structure in the center of low pressure”. In order to avoid confusion, it is necessary to point out that the anticyclonic structure refers only to vertical movements in the eye of the typhoon, i.e. upwards, as in the anticyclone!

Figure 9. It is variation trend of SSTA, not SST!

 Figures 10, 11 and 12. Define what applies to Figures a) to f)?

  1. GENERAL REMARK

The amounts of individual quantities and their units must be separate, e.g. 20.5 m / s.

Author Response

(The authors gave the same response as above.)

Round 2

Reviewer 1 Report

Statistical characteristics of the response of the South China Sea to westward typhoons

by Zhaoyue Ma1, Yuanzhi Zhang, Renhao Wu and Rong Na

The author of the manuscript have addressed many of my questions and comments and I find the revised version an improvement over the first one. However, there are still outstanding issues that I believe need to be addressed before the paper is published.

Below are my comments to the authors:

17 – obvious cooling of what? It should be specified

27 – typhoon forming on the ocean surface implies it is oceanic phenomena, maybe over the ocean surface would be better

97 – I see that the manuscript has been edited for English and although in general it is an improvement there instances where while English is correct I don’t think it expresses what the author what to say. In this sentence somehow ‘high evaporation’ from previous version was replaced by ‘consequent high’. The meaning of these two phrases is very different I I believe the it is the first one old one that actually applies here.

180 - The six data were compared by drawing. It is not clear to me what does this mean.

Specific remarks:

Title: the paper deals with the change in SST after a passage of a typhoon while the title suggested a responds of a much larger scope. It should probably be made more specific.Response: Thank you very much.We have revised the title as “Statistical characteristics of the response of sea surface temperatures to westward typhoons inSouth China Sea”.

Ok

Number of cases: why are only 30 typhoons considered, what is the total number and what is the total number of westward typhoons over the study periodResponse: Thank you very much.According to the statistics of the best typhoon pathprovided by the Tropical Cyclone Data Center of the China Meteorological Administration, a total of 518 typhoons occurred in the Northwest Pacific during the study period (1998-2018), but only 30 westbound typhoons passed through the South China Sea, so this article Only consider these 30 westbound typhoons.

Ok

66 –what about rain, radiative processesResponse: Thank you very much for your comment, but wefocused to consider the response of sea surface temperatures to westward typhoons inSouth China Searather than rain and radiative process.

Yes, but typhoon usually comes with rain and change to radiative fluxes. They also have an effect on SST specially very close to the surface which is the SST measured by satellite instruments. Why can these effects be neglected?

74 -moving speed of what?Response: Thank you very much. The corresponding revised at new line 74(Red word):At the same time, the decrease of sea surface temperature is also related to the pressure gradient and moving speed ofthetyphoon.

OK

88 -I don’t understand this sentenceResponse: Thank you very much. The corresponding revised at new line 88-90(Red word):If there is a strong cold vortex on the left side of the track, this will cause the typhoon’s left side to cool down more significantly than the right side[28].

OK

93-103 this process will only matter if it is a frequent occurrence but there is on mention here how often these conditions form, moreover since only 20 typhoons from the period of 20 years are considered here this leaves impression these conditions are not very frequent and would not make big impactResponse:

Thank you very much. The main purpose of this part is to explain the general process of the upper ocean’s response to typhoons and its significance to actual fishery production and ocean transportation. It is also applicable in the South China Sea.The corresponding revised at new line 97-108(Red word):After typhoon, the typhoon caused strong mixing of water bodies, which led to an in-crease in the energy and material exchange between the upper ocean and the subsur-face water, the SST decreased significantly, the depth of the mixed layer increased, and the sea surface chlorophyll concentration increased significantly SST decreased signif-icantly, mixed layer depth increased and sea surface chlorophyll concentration in-creased significantly [32-37].Which has a great impact on the economic and social effects of the local offshore fishery, marine transportation, marine operations and other production departments. Therefore, the response of the upper oceanto typhoons has been one of the research hotspots in recent years.References:Li, Y.X. Typhoons’ effects on ocean cyclonic eddies in the North Western Pacific. Master’s degree, University of Science and Technology of China, 2015.Zedler, S.E.; Dickey, T.D.; Doney, S. C.; et al. Analyses and simulations of the upper ocean's response to Hurricane Felix at the Bermuda Testbed Mooring site: 13–23 August 1995. J. Geophy. Res. Oce.,2002, 107(C12), 1–29.Black, P.G. Ocean temperature changes included by tropical cyclones. Dis. Abs. Int. 1983, 44, 1487.

I understand the significance of the event when it happens but my point here is that it doesn’t happen all that often. Or is once a year significant enough.

124 -assume it is 10.3 typhoons per year but it should be spelled out here

Response: Thank you very much.The corresponding revised at new line 120-133(Red word):It is the largest marginal sea on the east coast of Asia in the Western Pacific Ocean and one of the areas with the most frequent tropical cyclones and typhoons, with an aver-age of 10.3 typhoons passing throughperyear [48,49].

OK

149-159, this discussionof microwave SST data if very brief. It is the data set on which the whole study is based yet there is no discussion of the accuracy of the microwave SST, its limitations, why it was chosen above other data set (numerical analysis, drifter data, other satellite instruments). The problems with microwave SST in strong winds and rain –very much present in typhoons in particular would be a cause for concern. There is also the question of the skin SST measured by satellite instruments vs bulk SST more representative of the mix layer which I guess would be more appropriate for the purpose of this study.

Response: Thank you very much for your valuable comments. I am very sorry that the description of the SST data in the manuscript is too simple to cause you inconvenience and difficulty in understanding.First of all, I need to explain to you that the microwave SST data used in this article are the SST after the typhoon disappears and the SST before the typhoon is generated. Therefore, although the severe weather brought by the typhoon will have an impact on the microwave SST, the impact should be smaller relative to the impact during typhoon.In addition, the SST data used in this study is Optimally Interpolated (OI) SST daily products provided by Remote Sensing Systems (REMSS). This data set integrates SST data measured by multiple microwave radiometers, removes the influence of daytime temperature and the SST data polluted by rainwater, can better reflect the trend of sea surface temperature changes after the typhoon, and has been widely used.The corresponding revised at new line 163-185(Red word):

Traditional infrared and visible light remote sensing methods are susceptible to cloud, rain and other complex weather phenomena, making it difficult to observe and study the low-level structure of typhoons. Microwave remote sensing can be observed day and night, and it can penetrate clouds and fog with little influence, making its application in extreme weather such as typhoon conditions unmatched by other types of sensors[45]. This article uses Optimally Interpolated (OI) SST daily products (http://data.remss.com/) provided by Remote Sensing Systems (REMSS). The data in this data set before June 2002 (1998-2002) only includes Measuring Mis-sion (TRMM) Microwave Imager (TMI) data, the spatial span of the data data is 40°N-40°S, its time resolution is 1 day, the spatial resolution is 0.25°×0.25°, and SST data error is 0.12°C. The SST data set after June 2002 includes TMI, Advanced Microwave Scanning Radiometer for EOS (AMSR -E), The Advanced Microwave Scanning Radiometer 2 (AMSR-2), WindSat, or GPM (The Global Precipitation Measurement) Microwave Imager (GMI) daily SST data, its time resolution is 1d, spatial resolution The rate is 0.25°×0.25°; data covers the world, and the error of SST data is 0.01°C. This data set removes the influence of daytime sea surface temperature warming and the SST data of rainwater pollution, and meets the research requirements of sea surface temperature changes after the typhoon passes.References:Yang,X.X.; Tang, D.L. Location of sea surface temperature cooling induced by typhoon

in the South China Sea. J. Trop. Oceanogr.2010, 29, 26–31.Li, Y.X.Typhoons’ effects on ocean cyclonic eddies in the North Western Pacific.Master’s degree, University of Science and Technology of China,2015.Gentemann, C.L.; Meissner, T.; Wentz, F.J. Accuracy of satellite sea surface temperatures at 7 and 11 GHz, IEEE Transactions on Geoscience and Remote Sensing, 2010,48, 1009-1018.Wentz, F.J.;Gentemann,C.L.;Smith, D.K.et al.Satellite measurements of sea surface temperature through clouds, Science, 2000, 288, 847-850.

153 –the 0.01C is a standard deviation of what ?Response: Thank you very much.The corresponding revised at new line183(Red word):data covers the world, andthe error ofSST data is 0.01 °C.

I very much doubt this statement is correct. In fact the Remote Sensing Systems data the authors claim to use in this study contain a field that is estimate of error and it is at least 10 times the value given here and in average about 50 times the value.

160-166, meaning of the average temperature here is not clear, is this the average temperature of the whole basin? This needs to better defined.

Response: Thank you very much.The corresponding revised at newline 191-195(Red word):

In order to show the influence of Typhoon on the sea surface temperature of the upper ocean more intuitively, this paper deals with the temperature data as follows: First, in order toavoid abnormal SST in the single day before the typhoon, reduce the calculation error, the sea surface temperature data of the 2 days before the occurrence of the typhoon in the South China Sea (5°-25°N, 100°-140°E) is averaged to obtain the average sea surface temperature mean1 of the sea area.

181 –again is 30 westward typhoons the total number of westward typhoons between 1998 ad 2018, if not why these 30 were chosen

Response: Thank you very much.As mentioned earlier, a total of 30 westbound typhoons occurred in the study area from 1998 to 2018.

OK

Table 2: and throughout the paper the Maximum cooling rage is probably not the best term for this parameter as range implies two values (lower and upper limit) the maximum value of cooling is just one number.

Response: Thanks for the correction.The corresponding revised at new table2:" Maximum cooling rage " is replaced by " Maximum cooling value".

OK, however to follow up – my understanding is that the maximum cooling is the maximum value of cooling observed over the are of consideration up to 6 days after the passage of a typhoon – i.e. this is just one point on the map. Is that correct?

I am not sure if that indeed is the best value to use (grate potential for error). Or is it spacial average over certain domain (probably better). The error estimate of this value is needed. Also I understanding is that with few exceptions the maximum cooling occurs on day 1 but it still would worthwhile to have it in the table as the ‘odd’ cases would possibly standout more clearly.

187 –cooling rate has not been defined but typically implies change over time so no consistent with the units of ‘%’

Response: I'm very sorry that our unclear expression has caused your understanding difficulty.The corresponding revised at new line 225-227(Red word):

Through statistics, we can find that 20 of the 30 westward typhoons from 1998 to 2018 experienced significant cooling, the occurrence of cooling typhoons accounted for 66.67% of the total westbound typhoons.

OK

Figure 3, it seems that the cooling range=SSTA , just use SSTA and drop the cooling range. Also, the figure axes should probably be reversed as the minimum pressure and the max wind speed seem to the the independent variables that drive the SSTA.

OK

Figure 4 could use the position of the typhoon’s center along the track on each day. That would provide addition information about the position of the center of the cooling and the

center of the typhoon over time.

Response: Thank you very much.In fact, the research time of this article focuses on the typhoon's extinction, and mainly studies the changes in sea surface temperature after the westwardtyphoon, rather than the SST changes as the typhoon moves, so the real-time typhoon center is not shown in figure.

OK

Figure 5 could use a legend to identify the tend lines for each typhoon

Response: Thank you very much.The corresponding revised at new Figure 5:Figure 5. Rising trend line of SST after westward typhoon(The legend part is the typhoon number)

OK

Figure 7 -13, again it would be good to see where the center of the typhoon is in each panel. this is speciallyimportant in figure 13 which attempts to explain an interaction between two typhoons

Response: Thank you very much.The typhoon center has been added to the picture that needs special explanation.

Figure 8 and discussion, the snapshots in panels of Figure 8 should state the time after passage for each panel otherwise the claims made wrt TY Usagi in particular cannot be supported.

Response: Thank you very much.We have revised it in the manuscript.

OK

240 –I am not sure if there us a need for a new section here. Most of the ‘Discussion’ section seems to be really a continuation of the Results for the special cases.

Response: Thank you so much for your comments.We have revised it.

Figure 9, needs error bars (as do some of the earlier Figures too). Also putting tick labels

OK, so here we have error bars that are more what I would expect, so about 1 deg. This is very different the the previously stated SST error of 0.01 C. How is this error calculated?

Obviously, with an SSTA error of ~1C it is harder make statements regarding recover rate.

The error bar should also be added to the earlier Figures, so Figure 3, 5, 6 and 6.

273 –Idon’t see how the slope is ranging from 0.1 to 0.5

Response: Thank you very much.The corresponding revised at new line 339(Red word):The slope of the temperature rise curve was 0.2234, which is also between 0.1-0.5.

OK

278 -‘Some scholars’ should be replaced by names of those scholars. I guess we should infer that the presence of the cold vortex affected the SST response to the TY Lingling but if that’s the message then it should be clearly stated here.

Response: Thanks foryour comments.The corresponding revised at new line 344-350(Red word):Yang et al.studied the surface height data of the South China Sea and found that there was an obvious cold vortex near the typhoon track one week before the formation of typhoon Lingling. After the typhoon, the scope of the cold vortex expanded and the intensity wasgreatly strengthened [28].Therefore, we infer that the change in sea surface temperature after the typhoon passes by the cold vortex is different from that of other westbound typhoons. The cooling time is longer and the temperature recovery takes longer.

OK

286 –we cannot really see from the Figure when the path changed since there are no time stamps along the track so we don’t know when the TY is at any point in time. All tracks should indicate the position of TY center corresponding to the time of the panel

Response: Thank you very much.The typhoon center has been added to the picture that needs special explanation.

OK

305 -306, we cannot see the subtropical high in Figure 12 nor how it influenced the TY Parma. Authors could add a synoptic chart showing the subtropical high are just describe the synoptic situation but in current form the statement is incorrect. Again having location of the TY center in the panels would be helpful.

Response: Thank you very much.The reason why the subtropical high appears in the manuscript is because previous studies (He et al., 1995; Hu et al., 1995; He, 2000) have shown that the subtropical high is the main cause of changes in the typhoon path. In order to avoid ambiguity and improve the reading experience, this part of the manuscript has been revised.References:Hu, S. T.;Huang, D. W.;Zhang, R.L. Climatic Features of Sudden Changes of Tropical CycloneTrack over the Northern Area of South China Sea. Meteorological Monthly,1995,21(8):23-25.He, Z.; Hu, S. T.; Zhang, R.L. A Diagnosis-Predict Method of Sudden Change in Typhoon Track and Its Test Results in 1994. Meteorological Monthly,1995, 21(8):7-12.He, Z. On the forecasting technology of abrupt clockwise during tropical cyclones over the North South China Sea. ScientiaMeteorologica Sinica,2000,20(3):298-301.

The revised sentence still is not supported but Figure 12. The explanation given about is much better and should be added to the paper.

335 –352 the conclusion statements carry very little wight without any kind of error analysis.

Response: Thank you very much for your proposal. We have added the standard deviation analysis of SSTA and added the results to Table 2, "3.1. Statistics of the Influence of Westward Typhoon on Sea Surface Temperature" and Conclusion section.Responses

In my understanding the main parameter in the study is the maximum cooling value. So what is the accuracy of this estimate?

I am not sure what is the meaning of the meaning standard deviation in table 2. Is it SSTA over the entire domain, at what point of time (1,...6 days after passage). An explanation is need here.

Author Response

Dear Reviewer,

Thank you for your comments. We have revised and improved as suggested.

Best regards,

Yuanzhi
